# Acute Myeloid Leukemia: A Key Role of DGKα and DGKζ in Cell Viability

**DOI:** 10.3390/cells14211721

**Published:** 2025-11-01

**Authors:** Elisa Gorla, Marco Cristiano Cartella, Edoardo Borghetti, Ginevra Lovati, Luisa Racca, Teresa Gravina, Giorgio Biazzo, Gabriele Bonello, Valeria Malacarne, Veronica De Giorgis, Davide Corà, Marcello Manfredi, Alberto Massarotti, Andrea Graziani, Gianluca Baldanzi

**Affiliations:** 1Center for Translational Research on Autoimmune and Allergic Diseases, University of Eastern Piedmont, 28100 Novara, Italy; elisa.gorla@uniupo.it (E.G.); 20027794@studenti.uniupo.it (M.C.C.); 20033143@studenti.uniupo.it (E.B.); 20038356@studenti.uniupo.it (G.L.); luisa.racca@uniupo.it (L.R.); teresa.gravina@uniupo.it (T.G.); veronica.degiorgis@uniupo.it (V.D.G.); davide.cora@uniupo.it (D.C.); marcello.manfredi@uniupo.it (M.M.); 2Department of Translational Medicine, University of Eastern Piedmont, 28100 Novara, Italy; valeria.malacarne@unito.it (V.M.); andrea.graziani@unito.it (A.G.); 3Department of Molecular Biotechnology and Health Sciences, Molecular Biotechnology Center (MBC Guido Tarone), University of Turin, 10126 Turin, Italy; 4Department of Pharmaceutical Sciences, Università del Piemonte Orientale, 28100 Novara, Italy; alberto.massarotti@uniupo.it

**Keywords:** acute myeloid leukemia, diacylglycerol kinases inhibition, diacylglycerol kinase isoform silencing, drug sensitivity, proteome remodeling, lipid signaling

## Abstract

**Highlights:**

**What are the main findings?**
DGK inhibition reduces AML cell viability in an isoform- and cell line-specific manner, independently of DGK expression levels.Selective inhibition of DGKα or DGKζ causes distinct proteomic changes, leading to commonly downregulated pathways.

**What are the implications of the main findings?**
Therapeutic strategies targeting DGK should account for AML 
heterogeneity and isoform-specific effects.Effective DGK-targeted therapy will require biomarkers for patient stratification and combinatorial therapeutic approaches.

**Abstract:**

Acute myeloid leukemia (AML) is a heterogeneous disease with an unmet need for novel therapeutic drugs. Previous studies have reported the upregulation of diacylglycerol kinases (DGKs) in AML. This study investigated the effects of ritanserin, a DGKα-specific inhibitor, and DGKζ-IN4 or BAY 2965501, DGKζ-selective inhibitors, on a panel of AML cell lines. Ritanserin induced apoptotic cell death across all tested models, whereas DGKζ inhibitors triggered both apoptosis and necrosis to variable extents, with HL-60 cells being the most responsive to both compounds. Drug sensitivity did not correlate with DGKα or DGKζ expression levels, indicating that additional factors may influence cellular susceptibility. THP-1 proteomic profiling revealed that ritanserin broadly downregulated proteins involved in antigen presentation, cell cycle and metabolism, while BAY 2965501 affected a smaller and distinct but functionally similar protein subset, implying different mechanisms of action. Gene silencing confirmed AML cell line-specific dependence on DGK isoforms: HEL cells were sensitive to DGKα knockdown, HL-60 to DGKζ silencing, whereas K562 and THP-1 were resistant to both. These findings indicate that DGKs targeting can effectively reduce AML cell viability. However, AML heterogeneity and the limited selectivity of current inhibitors underscore the need for predictive biomarkers and combinatorial strategies to translate DGK inhibition into effective therapy.

## 1. Introduction

Acute myeloid leukemia (AML) is a heterogeneous group of clonal hematopoietic neoplasms characterized by the accumulation of myeloid blasts in the bone marrow and peripheral blood. This accumulation results from a blockade of differentiation combined with enhanced proliferative capacity of myeloid precursor cells, ultimately impairing normal hematopoiesis [1].

Despite recent advances in hematologic malignancy management, including the development of targeted and immune therapies and improved supportive care, AML outcomes remain poor. The 5-year relative survival is approximately 24%, with an incidence of 4.3 new cases per 100,000 individuals per year in the US [2]. Several factors contribute to this unfavorable prognosis, including its higher incidence in older adults, poor chemotherapy response, high relapse rates, and limited options for relapsed disease. AML also imposes a significant clinical and financial burden due to its heterogeneity in age and genetic profile, prolonged hospitalizations, and infection-related complications. Standard first-line treatment consists of intensive chemotherapy aimed at achieving remission, followed by additional chemotherapy, radiation therapy, or stem cell transplantation [3]. However, significant unmet clinical needs remain, such as the urgency for more effective and less toxic treatments, strategies to prevent relapse, and better support to improve patients’ quality of life [4].

Given the urgency to identify novel therapeutic strategies, particular attention has been directed toward the diacylglycerol kinase (DGK) family, especially the DGKα and DGKζ isoforms. DGKs comprise a group of ten different cytosolic or nuclear enzymes that catalyze the phosphorylation of the lipid second messenger diacylglycerol (DAG), generating phosphatidic acid (PA). Both DAG and PA are primarily localized to cellular membranes where these lipids facilitate the activation of several proteins, such as protein kinase C, RAS activators, the mammalian target of rapamycin, and various tyrosine kinases. Through these interactions, DAG and PA act as dynamic second messengers regulating proliferation, survival, motility, and metabolic activity [5,6,7]. Dysregulation of DAG/PA lipid signaling has been implicated in several human disorders, including cancer [8,9]. Among the ten DGK-encoding genes, DGKA and DGKZ, which code for DGKα and DGKζ proteins, respectively, are the most extensively investigated owing to their central involvement in regulating immune responses, cancer immunotolerance, and tumorigenesis [10,11].

Several lines of evidence point to a relevant role for DGKs in AML. Indeed, analyses of cancer databases, including The Cancer Genome Atlas, have revealed DGKα, DGKγ, DGKδ, DGKε, and DGKζ overexpression in AML, although without a clear correlation to specific subtypes. High expression of DGKα is associated with poor overall survival, whereas elevated DGKγ appears to be linked to a more favorable prognosis. Interestingly, the Beat-AML dataset showed that bone marrow cells express all DGK isoforms at very high levels, which often exceeded those observed in tumor samples, suggesting that DGKs expression may increase during hematopoietic differentiation [12]. Moreover, overexpression of DGKα in human bone marrow AML samples compared to normal samples, has been associated with poor clinical outcomes, suggesting a possible oncogenic role [12,13]. Among DGK isoforms, α and especially ζ exhibit frequent alterations in leukemic cell lines where they play distinct functional roles. For instance, in Kasumi-1 and KG-1α AML cell lines, treatment with the DGKα-specific inhibitor ritanserin resulted in significant inhibition of the phospholipase D–PA–sphingosine kinase 1 axis, suppression of the JAK–STAT and MAPK pathways, and downregulation of PARP, caspase-3, MCL1, and Bcl-xL. Ritanserin exerted cytotoxic effects on both cell lines, with half maximal inhibitory concentration (IC_50_) values of 51 μM and 38 μM, respectively, after 24 h, and 30 μM and 26 μM after 72 h. Similarly, treatment of bone marrow mononuclear cells from primary AML samples with 40 μM ritanserin induced apoptosis in up to 70% of cells, whereas apoptosis in normal hematopoietic stem cell samples remained below 30%.

Collectively, these findings suggest that ritanserin reduces AML cell viability through a complex mechanism involving DGKα inhibition, signaling pathway downregulation, and caspase-dependent apoptosis. Congruently, in xenograft mouse models injected with Kasumi-1, ritanserin administration led to a significant reduction in tumor burden and improved survival [13]. Conversely, in HEL and HL-60 cells, the DGKα-specific inhibitors CU-3 and AMB639752 did not significantly affect viability, whereas the ritanserin-related inhibitors R59022 and R59949 induced marked cytotoxicity after 24 h, with IC_50_ values of 32 μM and 72 μM, respectively, in HL-60 cells, and 49 μM and 80 μM, respectively, in HEL cells [12]. In K562 cells, DGKα translocates into the nucleus during both the G_1_ and G_2_ phases of the cell cycle, and its presence is required for efficient cell cycle progression. Inhibition or silencing of DGKα in these cells reduced proliferation and altered nuclear morphology, supporting a direct role for this isoform in mitotic regulation [14].

RAS-mutation in AML cells alters lipid metabolism and promotes DGKζ overexpression, suggesting an additional role for this isoform [15]. Consistently, knockdown of DGKZ via RNA interference in HL-60 cells induced significant apoptosis and G_2_/M phase arrest, accompanied by MAPK pathway downregulation, suppression of the anti-apoptotic protein survivin, and activation of caspase-3 and caspase-9 [16].

Given the persistent clinical challenges and biological heterogeneity of AML, this study aimed to expand upon the fragmented data present in the literature on the roles of DGKα and DGKζ in leukemic cell biology. To this end, two complementary approaches were employed: (i) the use of newly developed isoform-specific inhibitors; and (ii) transient siRNA-mediated silencing across a panel of four well-established, yet biologically distinct, cell line models relevant to AML research. This combined strategy enabled a comprehensive assessment of their effects on cell viability, apoptosis/necrosis, and global proteomic profiles, thus providing mechanistic insights extending beyond those reported so far.

To reflect AML heterogeneity, four distinct models were selected. HL-60 cells, derived from acute promyelocytic leukaemia [16], were chosen due to their strong differentiation potential along the myeloid lineage [17,18,19,20,21]. HEL cells, established from a case of erythroleukemia, were included for their erythroid characteristics and presence of relevant driver mutations, making them suitable for modeling pure erythroid leukaemia [22]. THP-1 cells, obtained from acute monocytic leukaemia, harbor the typical t (9;11) translocation and reproduce chemoresistant and invasive phenotypes [23]. Upon stimulation they can differentiate into macrophage-like cells [24], providing a model to investigate DGK functions in monocyte-to-macrophage transition. Finally, K562 cells, although originally derived from chronic myeloid leukemia (CML), were included because of their frequent use in AML research owing to their conserved myeloid properties and ability to differentiate towards erythroid or megakaryocytic lineages under specific treatments [25,26,27].

In the rapidly growing field of DGKα inhibitors, ritanserin was selected as a reference compound. Ritanserin is a synthetic compound initially developed as a selective antagonist of serotonin 5-HT_2_A and 5-HT_2_C receptors, with an inhibition constant (Ki) of 0.45 nM and 0.71 nM, respectively [28]. Although never approved for clinical use, it has been tested in clinical trials at doses ranging from 1 to 30 mg, and its favorable oral bioavailability and tolerability in early trials have maintained its relevance in research [29,30]. Besides its serotonergic activity, ritanserin has been identified as a potent and selective inhibitor of DGKα [31,32].

To evaluate the effects of DGKζ inhibition on AML cell viability, two selective inhibitors were used: DGKζ-IN-4 and BAY 2965501. DGKζ-IN-4, a heteroaryl carboxamide compound, has been reported as a potent and selective inhibitor of DGKζ with an IC_50_ of 0.4 nM while also increasing IL-2 release by T cells by 10-fold in the nanomolar range (Watanabe et al. Patent WO2022114164). BAY 2965501, another DGKζ-selective inhibitor, showed in vitro activity by increasing natural killer (NK)- and T-cell-mediated tumor cell killing and enhancing IL-2-induced NK cell activation. In syngeneic murine tumor models, oral administration of BAY 2965501 improved T cell efficacy, reactivated exhausted T cells, and reduced tumor growth [33,34]. Currently, BAY 2965501 is undergoing a first-in-human Phase I clinical trial to evaluate its safety, tolerability, pharmacokinetics, pharmacodynamics, best dosage and preliminary efficacy in patients with advanced solid tumors (Clinical trial ID NCT05614102 https://www.clinicaltrials.gov/study/NCT05614102, accessed on 20 September 2025).

## 2. Materials and Methods

All material used are listed in Appendix A.

### 2.1. Cell Lines and Culture Conditions

The HL-60 cell line was derived in 1976 from the peripheral blood of a 36-year-old woman diagnosed with acute promyelocytic leukemia [17]. HL-60 cells exhibit multipotentiality and can differentiate along various myeloid lineages: upon treatment with phorbol esters (DAG analogues), they differentiate into macrophage-like cells in a process counteracted by DGKγ [18,19], while exposure to vitamin D_3_ induces monocytic differentiation [20] and dimethyl sulfoxide (DMSO) promotes granulocyte differentiation in a DGKα-dependent manner [21].

The HEL cell line originated in 1980 from the peripheral blood of a 30-year-old man who experienced a relapse of erythroleukemia following treatment for Hodgkin lymphoma. HEL cells are positive for KIT, harbor TP53 mutations, and express high level of glycophorin A, making them a model for pure erythroid leukemia [22].

The THP-1 cell line was obtained from the peripheral blood of a 1-year-old male patient with acute monocytic leukemia. THP-1 cells serve as a model of this AML subtype, as they harbor the typical t (9;11) (p21.3;q23.3) translocation, which generates the AF9-MLL fusion protein that promotes expression of genes involved in epithelial–mesenchymal transition and confers chemoresistant and invasive phenotypes [23]. Upon stimulation with phorbol esters, THP-1 cells cease proliferation, become adherent, and differentiate into macrophage-like cells [24].

The K562 cell line was derived from the pleural effusion of a 53-year-old woman in the terminal blast crisis phase of CML. Although K562 cells presents the Philadelphia chromosome containing the BCR-ABL1 fusion gene, they are widely utilized in AML research due to their myeloid characteristics and their ability to differentiate toward erythroid or megakaryocytic lineages under specific treatments [25,26,27].

These four cell lines were employed in this study as in vitro models of AML, cell line identity was verified through a Cell Line Authentication Test conducted by Eurofins Genomics (Ebersberg, Germany) or by the internal university service. The resulting markers matched 100% with those reported in the ATCC database. Notably, an additional K562 clone, described in Appendix A and kindly provided by Prof. L. Manzoli (Bologna, Italy), is reported as K562-related, differing from K526 by a single marker. Cells were cultured in RPMI-1640 medium (ThermoFisher Scientific, Waltham, MA, USA) supplemented with 10% heat-inactivated fetal bovine serum (FBS) and 1% penicillin-streptomycin, following the handling protocols provided by ATCC (Manassas, VA, USA). Additionally, healthy donor blood samples were provided by the UPO biobank under approved Ethical Committee authorization (Comitato Etico Interaziendale A.O.U. Maggiore della Carità 865/CE-CE206/2022 date 24 October 2022) and with informed consent for research use. Peripheral blood mononuclear cells were isolated by density gradient centrifugation using Ficoll-Paque PLUS (GE 14 Healthcare, Chicago, IL, USA), washed, and resuspended at a concentration of 2 × 10^6^ cells/mL in RPMI-1640 medium containing 10% heat-inactivated FBS. Peripheral blood lymphocytes (PBLs) were selected by the treatment with 1 μg/mL anti-CD3 (clone OKT3) and anti-CD28 (clone CD28.2) human antibodies for 72 h. Activated T cells were then cultured in complete medium supplemented with 100 IU/mL recombinant human IL-2 (rhIL-2, PeproTech, Cranbury, NJ, USA) at a density of 0.5 × 10^6^ cells/mL.

### 2.2. Cell Viability Assays

The alamarBlue assay was performed to assess cell viability in presence of chemical inhibitors. A total of 50,000 cells per well were seeded in a 96-wells plate in a volume of 100 μL of complete medium together with 10 μL of 10× concentrated inhibitor solution and then incubated for either 24 or 72 h. Each experimental condition was performed in technical quadruplicates, using cells treated with equal amounts of DMSO vehicle as control and medium alone as background. After incubation, alamarBlue was added to each well at a final concentration of 0.15 mg/mL. The fluorescence of the samples was measured 24 h later using a Spark 10 M Multimode Plate Reader (Tecan, Switzerland) with excitation and emission wavelengths of 535 nm and 590 nm, respectively. For dose–response curve fitting, fluorescence values were normalized according to the following formula:*Viability* (%) = [(*sample mean fluorescence*)/(*DMSO control mean fluorescence*)] ∗ 100

Data are the results of at least five independent experiments analyzed with GraphPad Prism 10 as [inhibitor] vs. normalized response curve (variable slope, after manual outlier removal).

To determine the type of cell death, apoptotic and necrotic cells were detected by flow cytometry using the Apoptosis Detection Kit Annexin V-FITC (Invitrogen, Carlsbad, CA, USA). For each sample, 500,000 cells, pretreated for 24 h, were stained according to manufacturer’s instructions, using 7-aminoactinomycin D (7AAD) instead of propidium iodide. Samples were analyzed on a BD FACS Symphony A5 flow cytometer (Franklin Lakes, NJ, USA) using BD FACSDiva 9.1 software. Data were gated based on FSC-A vs. FSC-H to exclude cell doublets and FSC-A vs. SSC-A to exclude debris. Different populations were identified as: live (Annexin V− and 7AAD−), necrotic (Annexin− and 7AAD+), early apoptotic (Annexin V+ and 7AAD−) and late apoptotic (Annexin+ and 7AAD+). Data are shown as mean ± SD of at least three independent experiments analyzed by GraphPad Prism 10 (control vs. [inhibitor]).

To assess cell viability, trypan blue exclusion assays were performed on transfected HL-60, HEL, THP-1, and K562 cells. Trypan blue reagent was added to samples (1:1 *v*/*v*), and the number of live/dead cells was automatically recorded by the Automatic counter TC20 (Invitrogen). Each experiment was performed in triplicate. Results are reported as both the total number of live and dead cells from a representative experiment and as the percentage of cell viability averaged from three biological replicates. All data were graphed and analyzed using GraphPad Prism 8 through one-way ANOVA and unpaired *t*-test.

### 2.3. Silencing

To achieve DGKA and DGKZ silencing, HL-60, HEL, THP-1, and K562 cells were electroporated with 80 nM of siRNA using Neon™ 100 µL Kit (MPK10025, Invitrogen) with Neon™ electroporator (ThermoFisher Scientific, Waltham, MA, USA) under the conditions reported in Table 1. All siRNAs used are listed in the Appendix A.

After siRNA transfection, cells were resuspended in RPMI-1640 medium with 10% heat inactivated FBS without streptomycin and penicillin and incubated for 72 h at 37 °C in 5% CO_2_. Following incubation, silencing efficiency was evaluated for each experiment by RT-PCR and Western blot, after which cells were used for viability assays.

### 2.4. Quantitative Real-Time PCR

Transfected cells were lysed in 200 µL of nuclease-free water supplemented with 2% thioglycerol. Total RNA was extracted from lysates by Kingfisher automated nucleic acid extractor (Thermo Fisher Scientific, Waltham, MA, USA with the MVP_2WASH_200_FLEX protocol and the MagMAXTM Viral/Pathogen Nucleic Acid Isolation Kit (Thermo Fisher) and then quantified using NanoDrop (Thermo Fisher). Subsequently, equal amounts of RNA were retrotranscribed into cDNA using the High-Capacity cDNA Reverse Transcription Kit (Thermo Fisher) following the manufacturer’s instructions. The relative expression of DGK isoforms (DGKA Assay ID Hs00176278_m1, DGKZ Assay ID Hs05025727_m1) was assessed by quantitative real-time PCR (qRT-PCR) using TaqMan technology. Glucuronidase beta (GUSB Assay ID Hs00939627_m1) served as the internal reference gene. qRT-PCR was performed in a 384-well plate according to the TaqMan™ Fast Advanced Master Mix kit (Thermo Fisher), using 4 µL of mix and 1 µL of cDNA and run on a Bio-Rad CFX384^TM^ (Bio-Rad, Hercules, CA, USA) real-time PCR system. The amplification protocol consisted of an initial denaturation at 95 °C for 20 s, followed by 50 cycles of denaturation at 95 °C for 3 s and annealing/extension at 60 °C for 30 s. The qPCR reactions were performed in technical triplicates. Data were analyzed with Bio-Rad CFX Maestro 4.1.2433.1219 software, using GUSB as reference and normalizing DGKA and DGKZ expression to the corresponding siRNA control.

### 2.5. Western Blotting

Cells were lysed in buffer containing HEPES 25 mM, NaCl 150 mM, EDTA 5 mM EGTA 1 mM NP40 1%, and glycerol 10%, supplemented with sodium orthovanadate (1 mM, Thermo Fisher) and protease inhibitor cocktail (0.0089 mg/mL, Sigma Aldrich, St. Louis, MO, USA). Samples were incubated for 15 min at 4 °C, under gentle, constant shaking, and then centrifuged at 12,000× *g* for 15 min at 4 °C. The concentrations of proteins in lysates were quantified by Qubit Protein BR Assay Kit (A50669, Thermo Fisher Scientific, Waltham, MA, USA) according to the manufacturer’s instructions. Equal amounts of proteins were separated on SDS–PAGE gels and subsequently transferred onto polyvinylidene fluoride (PVDF) membranes. Membranes were then blocked with 3% (*w*/*v*) BSA for 1 h at room temperature and incubated overnight at 4 °C with the appropriate primary antibodies under gentle agitation. The following day, membranes were washed 3 times in TBS-T buffer and then incubated for 1 h at room temperature with HRP-conjugated rabbit or mouse secondary antibodies (1:5000) under gentle agitation. After three additional washes with TBS-T, the membranes were developed using the Western Chemiluminescence Substrate (PerkinElmer, Shelton, CT, USA) and visualized with the ChemiDoc™ Imaging System (Bio-Rad, Hercules, CA, USA). Band intensities were quantified by densitometry using Bio-Rad Image Lab 6.1 software, normalized to β-actin, and further normalized to the respective siRNA controls. Finally, results were analyzed and graphed using GraphPad Prism 8 with one-way ANOVA and unpaired *t*-test.

### 2.6. Simulations

All computational analyses were performed on a Tesla workstation equipped with two Intel Xeon X5650 2.67 GHz processors and running Ubuntu 20.04 (http://www.ubuntu.com). Preliminary information on the full-length human DGKα (735 amino acids, UniProt ID: P23743) and DGKζ (928 amino acids, UniProt ID: Q13574) sequences was obtained from the UniProt database (https://www.uniprot.org/) [35]. Since the complete resolved structures are not yet available, predicted models generated by AlphaFold [36,37] (AF-P23743-F1-v4 and AF-Q13574-F1-v4) were analyzed. The predicted protein structures were visualized using PyMOL (version 3.1.4.1), with structural domains annotated according to the Simple Modular Architecture Research Tool (SMART, version 9.0 https://smart.embl.de/, accessed on 5 March 2024) [38].

Potential ligand-binding pockets were then identified using FPocketWeb [39] (https://durrantlab.pitt.edu/fpocketweb/, accessed on 5 March 2024), focusing on putative pockets located near the catalytic sites of each isoform for further analysis. Molecular docking analyses were then performed using AutoDock Vina—Smina version 15 October 2019. based on AutoDock Vina 1.1.2. [40,41]. The docking procedure employed the PDB structure of each protein, the selected binding pocket as the target site, and a known inhibitor—provided as a SMILES strings—as ligands. The top-ranked docking poses corresponding to the inhibitor and the respective binding pocket were finally reassembled in PyMOL for structural visualization. In details, the ritanserin–pocket 56 complex was analyzed for DGKα, while the BAY 2965501 and DGKζ-IN-4 ligands were docked within pocket 65 of DGKζ.

### 2.7. Proteomics

Cells were lysed in RIPA buffer and sonicated. Proteins were precipitated with cold acetone, pelleted, and resuspended. Reduction was carried out in 25 µL of 100 mM NH_4_HCO_3_ with 2.5 µL of 200 mM DTT (Merck, St. Louis, MO, USA) at 60 °C for 45 min, followed by alkylation with 10 µL of 200 mM iodoacetamide (Merck, St. Louis, MO, USA) for 1 h at room temperature in the dark. Excess iodoacetamide was quenched by adding 200 mM DTT. Proteins were then digested with trypsin, dried by SpeedVac, and desalted. Peptides were analyzed on an Ultimate 3000 RSLCnano system coupled to an Orbitrap Exploris 480 equipped with FAIMS (all from Thermo Fisher Scientific). Samples were loaded onto a reversed-phase C18 column (15 cm × 75 µm i.d., Thermo Fisher Scientific) and eluted at 500 nL min^−1^ using a 6–95% solvent B gradient over 41 min, followed by 1 min of re-equilibration at 6% B. Data were acquired in positive mode with a spray voltage of 2500 V and FAIMS in standard resolution at a compensation voltage of −45 V. Data-independent acquisition (DIA) was performed with a precursor m/z range of 400–900, using 8 *m*/*z* isolation windows with 1 *m*/*z* overlap, HCD collision energy of 27%, Orbitrap resolution of 30,000, RF Lens 50%, normalized AGC target 1000, maximum injection time 25 ms, and 1 microscan. DIA data were processed with DIA-NN (v1.8.1) in library-free mode, applying deep-learning prediction for spectra, retention times, and ion mobility. Searches assumed Trypsin/P specificity, precursor charges 1–4, peptide lengths of 7–30 amino acids, and precursor *m*/*z* values of 400–900, allowing up to two missed cleavages. Carbamidomethylation of Cys was set as a fixed modification; Met oxidation was set as a variable modification (maximum two per peptide). The false discovery rate was maintained at 1%.

Normalized data from two independent proteomic experiments were obtained, each comparing a treatment condition to DMSO: (1) ritanserin vs. DMSO and (2) BAY 2965501 vs. DMSO. For each condition, three biological replicates were analyzed.

Protein abundance differences were calculated separately for each treatment relative to DMSO using a *t*-test, generating log_2_ fold change (log_2_FC) and adjusted *p*-values (*p*-adj). All proteins identified by UniProt ID were annotated with the corresponding gene symbol using QIAGEN Ingenuity Pathway Analysis (IPA; QIAGEN Inc., https://digitalinsights.qiagen.com/IPA, accessed on 5 June 2025). Proteins lacking a gene symbol were annotated as NA [42].

Data were filtered to include only proteins with |log_2_FC| > 2 and p-adj < 0.05, and retaining those annotated with a gene symbol. This resulted in 1210 differentially abundant proteins for the ritanserin vs. DMSO condition and 47 for BAY 2965501 vs. DMSO.

Volcano plots were generated for each treatment using the EnhancedVolcano package in R (https://bioconductor.org/packages/EnhancedVolcano, accessed on 12 June 2025), showing upregulated proteins in red and down-regulated proteins in blue, with labels for the ten most significantly modulated proteins. A Venn diagram was created to depict the overlap of differentially abundant proteins shared between the two treatments. These proteins, along with their gene symbols and log_2_FC values, were then used as input for IPA Core Analysis. The following filters were applied: human species and reference tissues/cells (hematopoietic progenitor cells, macrophages, monocytes, peripheral blood monocytes, and THP1 leukemic cells) [42]. The resulting Ingenuity Canonical Pathways were downloaded using standard filters and further refined to consider only those with a |Z-score| > 1. For ritanserin vs. DMSO, the top 15 most significant pathways were represented in a bar plot, while for BAY 2965501 vs. DMSO, only three significant pathways were identified and plotted.

## 3. Results

### 3.1. Ritanserin Is Cytotoxic and Induces Apoptosis in AML Cells

To further investigate the role of DGKα in AML, the analysis was extended beyond the data reported by Tan et al., in Kasumi-1 and KG-1α cells [13] to include additional AML cell models.

Initially, the potential interaction between DGKα and ritanserin was assessed, as this molecule is well known to inhibit selectively this isoform, although the structural bases of such inhibition are poorly characterized [43]. The predicted protein structure of DGKα was visualized (Figure 1A), and its functional domains were highlighted (Figure 1B). The most promising binding pocket was identified through molecular docking analysis, corresponding to pocket 56 (Figure 1C,D). These results are consistent with the findings by Mendez et al. [44].

Subsequently, cytotoxicity assays were conducted on THP-1, HL-60, HEL, and K562 cell lines. Cell viability was evaluated using the alamarBlue assay after exposure to ritanserin at concentrations ranging from 12.5 to 100 µM. Based on cell viability, dose–response curves at 24 and 72 h were generated (Figure 1E) and the IC_50_ values for each cell line were calculated. As expected, the IC_50_ values decreased across all tested cell lines with prolonged exposure times. THP-1 cells were the most sensitive to ritanserin (37.0 ± 1.4 µM after 24 h and 32.0 ± 1.5 µM after 72 h), followed by HL-60 (37.6 ± 1.5 µM after 24 h and 25.3 ± 0.9 µM after 72 h) and HEL (49.2 ± 2 µM after 24 h and 29.6 ± 1 µM after 72 h). K562 cells were the least sensitive (70.4 ± 2.2 µM after 24 h and 42.5 ± 1.8 µM after 72 h). PBLs from healthy donors, used as control, showed an IC_50_ of 42.9 ± 1.3 µM after 24 h and 32.5 ± 1.9 µM after 72 h (Appendix A). Apoptosis and necrosis assays were conducted at 24 h to evaluate the modality of cell death induced by ritanserin at concentrations near the IC_50_. Ritanserin treatment significantly increased apoptosis in all tested cell lines (Figure 1F, Appendix A). In particular, THP-1, HEL, and K562 cells predominantly underwent early apoptosis, while HL-60 cells exhibited late apoptosis. These findings confirm the dose- and time-dependent cytotoxicity of ritanserin in the micromolar range, consistent with previous reports in other AML models, while also revealing marked heterogeneity among cell lines and detectable toxicity toward untransformed cells at concentrations above 50 µM.

To rule out potential off-target effects of ritanserin, related to its known activity as serotonin 5-HT_2_A and 5-HT_2_C receptors antagonist, AML cells were treated with three additional serotonin receptors inhibitors: risperidone, metoclopramide, and altanserin. Risperidone is a second-generation antipsychotic with high affinity for 5-HT_2_A (Ki = 0.16 nM) and D_2_ (Ki = 3.13 nM) [45]. Metoclopramide is a dopamine D_2_ receptor antagonist with nanomolar affinity (Ki = 28.8 nM) and additional serotonergic activity, primarily used as an antiemetic [46]. Altanserin is a highly selective antagonist of the serotonin 5-HT_2_A receptor (Ki = 0.13 nM) that can be radiolabeled and employed as a radiotracer in PET neuroimaging studies [47].

These control compounds were tested using the same experimental protocol as described above for 24 h. The resulting dose–response curves (Appendix A) showed that serotonin receptor antagonists exerted negligible effects on AML cell viability, with the highest reduction in viability being only 11% following treatment with 100 μM metoclopramide. These findings support the conclusion that the cytotoxic effects of ritanserin in AML cell lines are unlikely to be attributed to interference with serotonergic signaling.

### 3.2. DGKζ-in-4 Has Heterogeneous Cytotoxic Effects and Induces Both Apoptosis and Necrosis in AML Cell Lines

To assess the impact of DGKζ inhibition on AML viability, DGKζ-IN-4 was tested using the same experimental procedure (Figure 2).

At first, the interaction between DGKζ-IN-4 and DGKζ was simulated. The predicted structure (Figure 2A) and domains (Figure 2B) were visualized, selecting pocket 65 for docking with DGKζ-IN-4 (Figure 2C,D). These findings align with those reported by Mendez et al. [44].

Concerning cytotoxicity, IC_50_ values decreased with longer treatment duration, similar to observations with ritanserin. HL-60 cells were the most sensitive to DGKζ-IN-4 after short exposure (21.2 ± 1.0 µM at 24 h and 19.9 ± 1.3 µM at 72 h), while THP-1 cells displayed the greatest sensitivity following prolonged exposure (31.3 ± 2.0 µM at 24 h and 12.4 ± 0.3 µM after 72 h). HEL cells were less responsive (50.7 ± 5.7 µM after 24 h and 20.7 ± 1.6 µM at 72 h), whereas K562 cells showed no detectable cytotoxic effects within the tested concentration range at 24 h and only mild sensitivity after extended exposure (32.9 ± 6.9 µM at 72 h). PBLs from healthy donors were sensitive to this drug with an IC_50_ of 28.7 ± 3.3 µM at 24 h and 15.1 ± 1.8 µM at 72 h (Appendix A).

Apoptosis and necrosis assays were conducted at 24 h using inhibitor concentrations close to the calculated IC_50_ values. K562 cells were not tested due to the lack of relevant cytotoxicity. DGKζ-IN-4 treatment led to a statistically significant increase in both apoptosis and, unexpectedly, necrosis across the tested AML cell lines (Figure 2F, Appendix A). Specifically, THP-1 and HL-60 cells underwent both late apoptosis and necrosis, while HEL cells showed early apoptosis accompanied by necrosis.

These findings demonstrate a dose- and time-dependent effect of DGKζ-IN-4 in the micromolar range, characterized by substantial variability among AML cell lines and the presence of resistant phenotypes. Moreover, untransformed PBLs were highly sensitive to this compound. In addition to the previously reported pro-apoptotic effects of DGKZ knockdown in HL-60 [16], these results indicate that chemical inhibition of DGKζ activity induces not only apoptosis but also necrosis.

### 3.3. BAY 2965501 Displays Highly Heterogeneous Cytotoxic Effects and Induces Both Apoptosis and Necrosis in AML Cell Lines

To further investigate the effects of DGKζ chemical inhibition in AML, a second specific inhibitor, BAY 2965501, was tested. Cytotoxicity assays were conducted following the previously described protocol (Figure 3A). As observed with DGKζ-IN-4, IC_50_ values decreased with longer exposure times. HL-60 cells were the most sensitive to BAY 2965501 (40.7 ± 4.4 µM at 24 h and 34.1 ± 3.0 µM at 72 h), followed by THP-1 cells (67.0 ± 3.5 µM after 24 h and 38.0 ± 3.0 µM at 72 h). HEL cells were less sensitive (92.0 ± 5.0 µM at 24 h and 58.1 ± 5.8 µM at 72 h), whereas K562 cells showed no detectable cytotoxic effects within the tested concentration range at 24 h and only minimal sensitivity after prolonged exposure (100 ± 3.2 µM at 72 h). PBLs from healthy donors showed an IC_50_ of 77.1 ± 4.8 µM after 24 h and 35.2 ± 0.7 µM at 72 h (Appendix A).

Apoptosis and necrosis assays were performed at 24 h using inhibitor concentrations close to the calculated IC_50_ values. K562 and HEL cells were excluded from these analyses due to the absence of relevant cytotoxicity. BAY 2965501 treatment induced a statistically significant increase in both apoptosis and necrosis across all the tested AML cell lines (Figure 3B, Appendix A). In particular, THP-1 cells primarily underwent early apoptosis, while HL-60 cells exhibited both late apoptosis and necrosis. Also in this case, docking simulations confirmed stable binding of BAY 2965501 within pocket 65 of DGKζ (Figure 3C).

Collectively, these findings confirm that BAY 2965501 exerts dose- and time-dependent cytotoxic effects in the micromolar range with substantial variability across AML cell lines and the presence of insensitive cell lines. PBLs showed only modest sensitivity. Consistent with the effects observed for DGKζ-IN-4, BAY 2965501 induced both apoptotic and necrotic cell death.

### 3.4. Correlation Between DGK Isoforms Expression and Sensitivity to Isoform-Specific Inhibitors

To determine whether sensitivity to DGK inhibitors was influenced by target expression, DGKA and DGKZ relative expression levels were assessed at both the mRNA and protein levels using q-RT-PCR and Western Blot, respectively. DGKA was most highly expressed in HEL cells, followed by K562 and PBLs, whereas HL-60 and THP-1 showed low expression at both mRNA and protein levels. Conversely DGKZ protein levels were highest in HL-60, lower in K562, HEL, and PBLs, and very low in THP-1. mRNA abundance differed slightly, with high expression in THP-1, intermediate levels in HEL and HL-60, and low levels in K562. The expression data were then correlated with the IC_50_ values of inhibitors targeting either DGKA (ritanserin) or DGKZ (DGKZ-IN4 and BAY 2965501) across AML cell lines and PBLs (Appendix A).

A weak correlation was observed between DGKA protein expression and ritanserin IC_50_ at both 24 h (R^2^ = 0.1557) and 72 h (R^2^ = 0.0021). Similarly, DGKA mRNA levels showed only a minimal association with ritanserin IC_50_ (24 h R^2^ = 0.0151; 72 h R^2^ = 0.0539), suggesting that DGKA expression has a limited predictive value for sensitivity to ritanserin.

The correlations between DGKZ expression and sensitivity to DGKZ-IN-4 were even weaker. DGKZ protein levels showed a moderate inverse correlation with DGKZ-IN-4 IC50 at 24 h (R^2^ = 0.3064) and no correlation at 72 h (R^2^ = 0.0149). In contrast, DGKZ mRNA levels displayed a stronger negative correlation at both time points (24 h R^2^ = 0.04248; 72 h R^2^ = 0.7963), indicating that sensitivity to DGKZ-IN-4 does not depend on high DGKZ expression.

A similar trend was observed for BAY 2965501. DGKZ protein expression was strongly inversely correlated with BAY 2965501 IC_50_ at both 24 h (R^2^ = 0.5884) and 72 h (R^2^ = 0.09594), while mRNA levels exhibited a slightly weaker yet consistent negative association (24 h R^2^ = 0.1267; 72 h R^2^ = 0.7992).

Altogether, these results indicate that, at least in AML cell lines, sensitivity to isoform-specific DGK inhibitors is not predicted by the expression levels of their corresponding targets.

### 3.5. Mechanism of Action of DGK Inhibitors

To understand the mechanism by which DGKα and DGKζ inhibitors decrease cell viability, the effects of ritanserin and BAY 2965501 on the proteome of THP-1 cells were investigated.

For this purpose, THP-1 cells were treated for 24 h with ritanserin (50 µM) and BAY 2965501 (75 µM), concentrations close to their respective IC_50_ values. Cells treated with equal amounts of DMSO (vehicle) were used as controls.

Protein extraction and quantification by mass spectrometry revealed that, in ritanserin-treated samples, among the 5772 proteins detected, approximately 1200 were significantly downregulated, while only 14 were significantly upregulated (increased by at least two-fold or at least halved, with an adjusted *p*-value < 0.05, Figure 4A and Appendix A). The predominance of downregulated proteins is consistent with a cell population undergoing growth arrest and apoptosis (Figure 1E). In line with the findings of Tan et al., PARP, Caspase3, MAPK, JAK and STAT proteins were significantly downregulated [13]. The most upregulated proteins (Figure 4A) were associated with oxidative stress, extracellular matrix remodeling, nuclear integrity and homeostasis, innate immune activation, cell adhesion, lipid metabolism, and possibly Notch pathway modulation.

Pathway enrichment analysis indicated that ritanserin affected more than 500 pathways, mainly suppressing those involved in MHC I-mediated antigen processing and presentation, TCR signaling, cellular metabolism, cell cycle progression, DNA/RNA synthesis, protein homeostasis, oxidative stress response, and nucleotide excision repair (Figure 4D).

In comparison, BAY 2965501 treatment resulted in a markedly lower number of downmodulated proteins (47; Figure 4B and Appendix A), despite a similar total of quantified proteins (5806), suggesting a more limited impact on the proteome. Notably, the set of proteins modulated by BAY 2965501 differed substantially from those affected by ritanserin, with only 11 shared proteins (Figure 4D), indicating distinct effects on the cellular proteome. Due to the minor number of affected proteins, the number of significatively downmodulated pathways is reduced to three: (i) the mitotic G_2_-G_2_/M phases; (ii) the protein ubiquitination pathway; and (iii) class I MHC-mediated antigen processing and presentation (Figure 4E). These pathways are associated with altered protein degradation, vesicular trafficking—including modulation of several Ran and Rab regulators—and reduced cell proliferation. Interestingly, all three were also among the top 15 ritanserin-regulated pathways, suggesting that, although the two DGK inhibitors modulate distinct protein subsets, they converge on similar biological processes.

### 3.6. Role of DGKα and DGKζ in HL-60 Viability

DGKs have both enzymatic and scaffolding functions in cell biology [48,49]. To complement the data obtained with chemical inhibitors and overcome specificity issues, DGK α and DGKζ were silenced in the AML cell line panel.

In HL-60 cells, DGKα and DGKζ mRNA and protein levels were quantified at 72 h after knockdown using siRNAs. DGKA mRNA levels were reduced by approximately 40% with siRNA1 and 20% with siRNA2, while DGKα protein expression decreased by about 50% with both siRNAs (Figure 5A). For DGKZ, mRNA expressions decreased by approximately 55% (Figure 5C).

To assess the biological effects of DGKα and DGKζ reduction on HL-60 cell viability, trypan blue assays were performed 72 h post-transfection. As shown in Figure 5B (left panel), representing the mean percentage of live cells from four independent experiments, viability was reduced by about 20% in HL-60 cells transfected with DGKA siRNA1 but was unaffected by siRNA2. The corresponding absolute cell count from a representative experiment (Figure 5B, right panel) indicated that both control and DGKA-silenced cells showed a net increase in cell number relative to the initial seeding density (dotted line). On the other hand, a modest non-significant reduction in proliferative capacity was observed in cells transfected with DGKA siRNA1, with no difference detected in those transfected with siRNA2. In contrast, DGKZ knockdown resulted in a statistically significant reduction in mean live cell percentage across seven experiments, with an average decrease of approximately 37% (Figure 5D, left panel). The absolute cell count of a representative experiment (Figure 5D, right panel) showed that only control cells displayed a net increase relative to the initial seeding density (dotted line), whereas DGKZ-silenced cells remained nearly unchanged in number, indicating a block in proliferation or increased apoptosis (Figure 5D, left panel).

To further investigate the induction of cell death upon DGKZ silencing, apoptosis and necrosis were assessed. As shown in Figure 5E and Appendix A, DGKZ silencing increased total cell death from 2.2% to 3.8%, primarily attributable to late apoptosis (*p* < 0.05), with minimal contributions from early apoptosis and negligible necrosis in both siRNA control and DGKZ-silenced cells.

Collectively, these results demonstrate that in HL-60 cells, DGKζ plays a critical role in sustaining cell viability. Its downregulation promotes a strong cytostatic effect accompanied by apoptotic cell death, whereas DGKα appears to be dispensable for viability under the conditions tested.

### 3.7. Role of DGKα and DGKζ in HEL Viability

Given that pharmacological inhibition with ritanserin, DGKζ-IN-4 and BAY 2965501 showed cytotoxic effects on HEL cells (Figure 1, Figure 2 and Figure 3), the same experimental approach was applied to investigate the contribution of DGKα and ζ isoforms in this cell line.

After 72 h, DGKA mRNA levels were reduced by approximately 63% with siRNA1 and 57% with siRNA2, as determined by qRT-PCR. At the protein level, DGKα decreased with both siRNAs, with a more pronounced effect observed with siRNA2 (Figure 6A). Similarly, DGKZ silencing led to an mRNA reduction of approximately 61%, which correlated with a marked decrease in DGKζ protein levels (Figure 6D).

To evaluate the biological effects of reducing DGKα and DGKζ protein on HEL cell viability, trypan blue assays were performed at 72 h. As shown in Figure 5B, DGKA silencing resulted in a significant reduction in the mean percentage of live cells across three independent experiments, corresponding to a 44% decrease with siRNA1 and 32% with siRNA2. The relative absolute cell count from a representative experiment revealed that only control cells exhibited a net increase in cell density compared to the initial seeding level (dotted line), whereas DGKA-silenced cells remained unchanged in number, indicating a proliferation block or increased apoptosis (Figure 6B, right panel).

To further examine the potential induction of cell death upon DGKA silencing, apoptosis and necrosis were investigated. As shown in Figure 5C and Appendix A, DGKA silencing further elevated total cell death from 6.2% to 11.4% with siRNA1 and to 7.5% with siRNA2. These increases were predominantly due to late apoptosis (*p* < 0.05), with negligible contributions from early apoptosis and necrosis. Conversely, DGKZ silencing did not significantly affect the mean percentage of live cells across three experiments (Figure 6E). The absolute cell count of a representative experiment indicated that both control and DGKZ-silenced cells showed comparable increases relative to the initial seeding density (Figure 6E, right panel), implying that DGKZ does not play a major role in regulating HEL cell survival or proliferation.

Taken all together, these findings demonstrate that in HEL cells, DGKα plays a critical role in maintaining cell viability, and that its downregulation promotes a strong cytostatic effect coupled with apoptotic cell death. In contrast, DGKζ appears to be dispensable for viability under the conditions tested.

### 3.8. Role of DGKα and DGKζ on THP-1 Viability

The specific contributions of DGKα and ζ isoforms to THP-1 cell viability were investigated using the same experimental approach. After 72 h, DGKA mRNA levels were reduced by 30% with siRNA1 and 32% with siRNA2, while DGKα protein expression decreased by 63% and 60%, respectively (Figure 7A). For DGKZ, a 50% reduction in mRNA and a 79% decrease in protein levels were observed (Figure 7C). As shown in Figure 7B (left panel), the percentage of viable cells from four experiments remained unchanged in THP-1 cells transfected with both DGKA siRNAs compared to control cells. The right panel of the same figure illustrates the absolute cell counts, revealing that both control and DGKA-silenced cells display an overall increase in cell number relative to the initial seeding density (dotted line). Likewise, after DGKZ knockdown, only a slight, not statistically significant, reduction in cell viability was detected compared to control cells (Figure 7D, left panel). The absolute cell counts (Figure 7D, right panel) confirmed that both control and DGKZ-silenced cells exhibited a net increase in cell number relative to the initial seeding density (dotted line). An apoptosis/necrosis assay was also performed to assess whether DGKA or DGKZ silencing induced significant cell death. As shown in Figure 7E and Appendix A, DGKA and DGKZ silencing minimally enhanced apoptosis in THP-1 cells compared to control cells, whereas necrosis remained negligible under all conditions. Thus, in THP-1 cells, decreasing DGKα and DGKζ protein levels does not significantly impair cell viability, suggesting a nonessential or functionally redundant role of these isoforms in this cellular context.

### 3.9. Role of DGKα and DGKζ in K562 Viability

The functional roles of DGKα and DGKζ isoforms were next examined in K562 cells. After 72 h, DGKA mRNA levels were reduced by 20% with siRNA1 and 45% with siRNA2, while DGKα protein expression decreased by 60% and 70%, respectively (Figure 8A). For DGKZ, mRNA levels decreased by 60%, accompanied by a 70% reduction in protein expression (Figure 8C). As illustrated in Figure 8B (left panel), the percentage of viable cells remained unchanged across three independent experiments in K562 cells transfected with either siRNA compared to control cells. The right panel of Figure 8B shows the absolute cell counts, demonstrating that both control and DGKA-silenced cells exhibited an overall increase in cell number relative to the initial seeding density (dotted line). Likewise, following DGKZ knockdown, silenced and control cells displayed comparable viability (Figure 8D, left panel). The corresponding absolute cell counts (Figure 8D, right) indicated that both control and DGKZ-silenced cells exhibited a marked increase in cell number relative to the initial seeding density (dotted line). Thus, in fast-growing K562 cells, decreasing DGKα and DGKζ protein levels does not result in a reduction in cell viability. This suggests a non-essential or redundant role in cell viability of those two isoforms in maintaining proliferation under the tested conditions.

Our findings are in contrast with the results reported by Poli et al. [14] regarding DGKA silencing. To address this discrepancy, the same K562 subclone used in that study was also examined. In these cells, DGKA silencing efficiency reached 50% with siRNA1 and 40% with siRNA2, while DGKZ silencing achieved a higher reduction (63% decrease in mRNA). Cell viability assessed by trypan blue exclusion confirmed that DGKA silencing had no detectable effect on viability or proliferation, whereas DGKZ silencing led to a modest (20%) reduction in both parameters compared to control cells (Appendix A).

## 4. Discussion

AML remains a major therapeutic challenge, highlighting the need for novel, effective treatments and the identification of new molecular targets. DGKα and DGKζ play critical roles in regulating a number of tumorigenic processes as well as in modulating the immune response to tumors [50,51,52,53,54]. Inhibition of these isoforms has been shown to induce antitumor effects in AML cell models, including reduced cell viability, cell cycle arrest, induction of apoptosis, and promotion of differentiation [12,13,14,16]. In this study, four representative cell line models relevant to AML research (i.e., HL-60, HEL, THP-1 and K562) were selected to determine the relevance of DGKα and DGKζ in AML cell viability using both isoform-specific chemical inhibitors and siRNA-mediated gene silencing.

R59949, R59022, and ritanserin constitute a family of structurally related DGKα specific inhibitors that bind within a pocket formed by the C1 and catalytic domains [32,55]. These inhibitors have been shown to decrease AML cell viability both in vitro and in vivo [12,13]. Here, a binding mode for ritanserin within the DGKα catalytic domain was proposed based on molecular simulations, and cytotoxicity experiments were extended to additional AML cell models. Consistent with previous studies, ritanserin reduced cell viability across all selected AML models tested, with THP-1 being the most sensitive (IC_50_ ≈ 37 μM), followed by HL-60 (IC_50_ ≈ 38 μM) and HEL (IC_50_ ≈ 49 μM), while K562 cells were more resistant (IC_50_ ≈ 70 μM). Additionally, the effects of ritanserin increased with prolonged exposure, showing clear dose- and time-dependent toxicity characterized by a significant increase in apoptosis without evidence of necrosis. Notably, untransformed but IL-2 stimulated proliferating PBLs exhibited complete cell death at a ritanserin concentration of 50 μM, consistent with the known role of DGKα downstream of IL-2 signaling [56]. These observations suggest that ritanserin cytotoxicity is not restricted to AML cells and that, at concentrations required to induce AML cell death, it may exert broader toxic effects in proliferating cells [57].

The mechanisms underlying ritanserin-induced cell death in AML are only partially understood. Previous studies have shown that ritanserin downregulates key signaling molecules, including sphingosine kinase 1, JAK–STAT, MAPK, PARP, caspase-3, MCL1, and Bcl-xL in Kasumi-1 and KG-1α (AML) cell lines [13]. In this study, proteomic profiling of THP-1 cells treated with ritanserin at near-IC_50_ concentrations confirmed a global downregulation of proteins, such as PARP, caspase-3, MAPK, JAK, and STAT, consistent with previous findings [13]. Functional enrichment analysis revealed that ritanserin treatment led to suppression of pathways involved in MHC I-mediated antigen processing and presentation, cell metabolism, cell cycle progression, DNA synthesis, and nucleotide excision repair. These alterations are in line with the molecular signatures of growth-arrested and apoptotic cells.

Since ritanserin is also known as a potent serotonin receptors antagonist, AML cells were treated with increasing amounts of altanserin, risperidone, and metoclopramide, three selective and strong serotonin/dopamine receptors antagonists. These compounds produced negligible effects on AML cell viability, with the greatest reduction (11%) observed following treatment with 100 μM metoclopramide. Therefore, ritanserin-induced cytotoxicity is unlikely to be mediated through serotonergic signaling, supporting a specific effect on DGKα inhibition.

To specifically target DGKζ activity, newly developed isoform-specific inhibitors recently reported in patents were employed. DGKζ-IN-4 is a selective inhibitor of DGKζ active in the nanomolar range and proposed as a potential therapeutic agent for cancer associated with immune cell activation or resistant to anti-PD-1 antibody/anti-PD-L1 antibody therapy (Watanabe et al. Patent WO2022114164). Similarly to ritanserin, DGKζ-IN-4 reduced cell viability in some of the selected AML models, with HL-60 being the most sensitive (IC_50_ = 21 μM), followed by THP-1 (IC_50_ = 31 μM) and HEL (IC_50_ = 51 μM), while K562 showed no detectable cytotoxic effects across the tested concentration range at 24 h and mild sensitivity after prolonged exposure (IC_50_ = 33 μM at 72 h). These results are the first available data on a DGKζ inhibitor in AML cell lines and support its potential application in this tumor type. However, PBLs from healthy donors were also sensitive to this drug with an IC_50_ of 29 μM after 24 h of exposure, indicating potential general toxicity. Apoptosis and necrosis assays revealed that DGKζ-IN-4 treatment induced a statistically significant increase in both apoptotic and, unexpectedly, necrotic cell death.

As this inhibitor is still poorly characterized, a second molecule, BAY 2965501, was also evaluated. This compound is known to inhibit DGKζ in vitro with an IC_50_ of 76 nM, as judged by ADP-Glo assays [58], and is reported to be highly selective for DGKζ. It is currently undergoing a phase I clinical trial in patients with advanced non-small cell lung cancer, gastric and esophagogastric junction adenocarcinoma, clear cell renal cell carcinoma, and melanoma [33,34] (Clinical trial ID NCT05614102 https://www.clinicaltrials.gov/study/NCT05614102, accessed on 20 September 2025). BAY 2965501 reduced cell viability in a subset of AML models with HL-60 being the most sensitive (IC_50_ of ≈41 μM), followed by THP-1 (IC_50_ of ≈67 μM), whereas HEL and K562 cells were resistant to this drug, with minimal cytotoxic effects. PBLs from healthy donor showed IC_50_ values of 77.1 ± 4.8 μM at 24 h and 35.2 ± 0.7 μM at 72 h (Figure 2), indicating that this inhibitor also affects untransformed cells at concentrations required to induce AML cytotoxicity. Apoptosis and necrosis assays confirmed that BAY 2965501 treatment induced a statistically significant increase in both apoptotic and necrotic cell death across sensitive AML cell lines, similarly to what observed with DGKζ-IN-4. Interestingly, molecular docking and dynamics simulations indicated that, despite their distinct chemical structures, DGKζ-IN-4 and BAY 2965501 may bind to the same catalytic domain pocket of DGKζ (Figure 2 and Figure 3). In future, these data could be complemented by the establishment of cell lines resistant to DGK inhibitors to study the potential resistance mechanisms.

To investigate the pathways modulated by BAY 2965501, THP-1 cells were treated with concentrations near to the estimated IC_50_, and the effects on the cellular proteome were analyzed. A limited number of proteins (47) were downregulated, comprising small G-protein regulators and proteins involved in protein trafficking and turnover (Figure 4B). These changes mainly affected three biological pathways: (i) the mitotic G_2_-G_2_/M phases; (ii) the protein ubiquitination pathway; and (iii) class I MHC mediated antigen processing and presentation. Even though most of the regulated proteins differed between ritanserin and BAY 2965501 treatments, these three pathways were among the most significantly modulated by both inhibitors, indicating that DGKα and DGKζ inhibition influences similar biological functions through different pathways.

The sensitivity to DGK inhibitors varied considerably among AML cell lines, highlighting the need to identify predictive biomarkers that could guide patient selection. Correlation analyses between IC_50_ values and DGKs expression at both mRNA and protein levels revealed no significant direct associations, indicating that DGK isoforms abundance is not the main determinant of inhibitor sensitivity and cannot be used for patient-tailored therapy. In previous reports [12], DGK isoforms overexpression was not associated with specific AML subtypes, although it may be associated with the occurrence of specific mutations, such as those leading to aberrant Ras pathway activation [15]. These findings underscore the need for further studies to clarify the role of DGKs in a heterogeneous disease like AML.

In agreement with earlier studies [12,13,14], DGK inhibitors effectively decreased AML cell viability in the high micromolar range. However, it will be necessary to extend this study to patient samples which could provide important insights into the roles of these enzymes in malignant cells compared to untransformed controls. Indeed, IL-2 stimulated proliferating PBLs also showed reduced viability, suggesting that the concentrations required to trigger AML cytotoxicity may also affect untransformed cells, limiting their potential therapeutic use. Both DGKα and DGKζ inhibitors are known to enhance immune responses [50,52], thus achieving a dual effect: direct tumor growth suppression and stimulation of immunosurveillance [10,11]. Considering the complexity and immunological activity of the tumor microenvironment, this dual action may lead to improved therapeutic efficacy at lower doses in vivo. Whether isoform-specific inhibition is advantageous in this setting or whether dual DGKα/ζ inhibition might be more beneficial remains to be determined [12,51]. Moreover, these molecules could be explored in combination with existing chemotherapeutic regimens or as adjuvants to improve the efficacy and persistence of CAR-T cells therapies, which are already approved for acute lymphoblastic leukemia and currently under clinical investigation in AML [59]. In addition the development of an efficient delivery system to enhance compound uptake in tumor cells could improve their efficacy.

When specifically addressing the role of DGKα and DGKζ through transient silencing, their relevance proved to be strongly cell line-dependent. DGKζ played a unique and crucial role in HL-60 cell viability as its knockdown led to a profound reduction in proliferation and an increase in late apoptosis, in line with data from Li et al. [16]. In the same model, DGKA silencing induced only a modest reduction in viability with one of the siRNA tested, suggesting that DGKα is less relevant for viability in this cell line.

In HEL cells, the opposite pattern was observed: DGKA silencing resulted in a significant reduction in cell proliferation and a reduction in viability. In contrast, DGKZ silencing, despite achieving substantial mRNA and protein downregulation, did not affect cell proliferation and viability.

In THP-1 cells, silencing of DGKA and DGKZ did not lead to significant changes in cell viability or proliferation. Similarly, K562 cells appeared to be quite resistant to both the effect of DGK inhibitors and DGKα and DGKζ gene silencing, showing no significant reduction in viability and proliferation. This latter finding is partially in disagreement with the results reported by Poli et al. [14] who, in addition to observing decreased proliferation following DGKα inhibitor treatment in the micromolar range, also reported reduced viability upon DGKα silencing. When the same K562 subclone used in that study was tested, a modest but significant decrease in viability was observed upon DGKζ silencing, whereas DGKA silencing again produced no detectable effect (Appendix A). The discrepancy may be attributed to differences in silencing efficiency, as DGKA mRNA levels in our experiments were reduced by only 50%. Indeed, using transient siRNA mediated silencing by electroporation allowed us to overcome long-term cell line adaptation to DGK depletion, which represents a limitation of our work due to the residual target expression observed across all experiments. In addition, the greater cytotoxicity observed with small-molecule inhibitors compared with siRNA knockdown may partly reflect off-target effects.

Overall, our findings indicate that AML cell viability depends variably on DGKα or DGKζ expression. While some AML lines, such as HEL (Figure 6) and previously described Kasumi-1 and KG-1α [13], rely primarily on DGKα, others, such as HL-60 (Figure 5 and [16]) and the K562 subclone (Appendix A) are DGKζ-dependent. Intriguingly, certain cell lines may not rely on either isoform, suggesting functional redundancy. To test this hypothesis and eventually determine the extent of this redundancy, further studies targeting additional DGK isoforms and employing combined inhibition or silencing approaches will be necessary. Furthermore, additional research on patient-derived AML blasts will be crucial to ascertain whether isoform dependency is associated with specific AML subtypes, molecular alterations, or stages of myeloid differentiation. Such studies should enable patient stratification and design optimization of future clinical trials.

## 5. Conclusions

Targeting DGKα and DGKζ effectively reduces AML cell viability, with both isoforms exhibiting distinct and cell type-specific functional roles. However, the observed drug sensitivity does not correlate with DGKα or DGKζ expression levels, indicating that additional cellular or molecular factors may modulate responses to these inhibitors. The substantial heterogeneity of AML, together with the limited selectivity of current DGK inhibitors, underscores the importance of identifying predictive biomarkers of sensitivity to DGKα or DGKζ blockade, which will be essential for defining responsive patient subgroups and guiding future clinical applications. Further studies employing patient-derived AML samples and in vivo models will be crucial to translate DGK inhibition into an effective therapeutic strategy.

## Figures and Tables

**Figure 1 cells-14-01721-f001:**
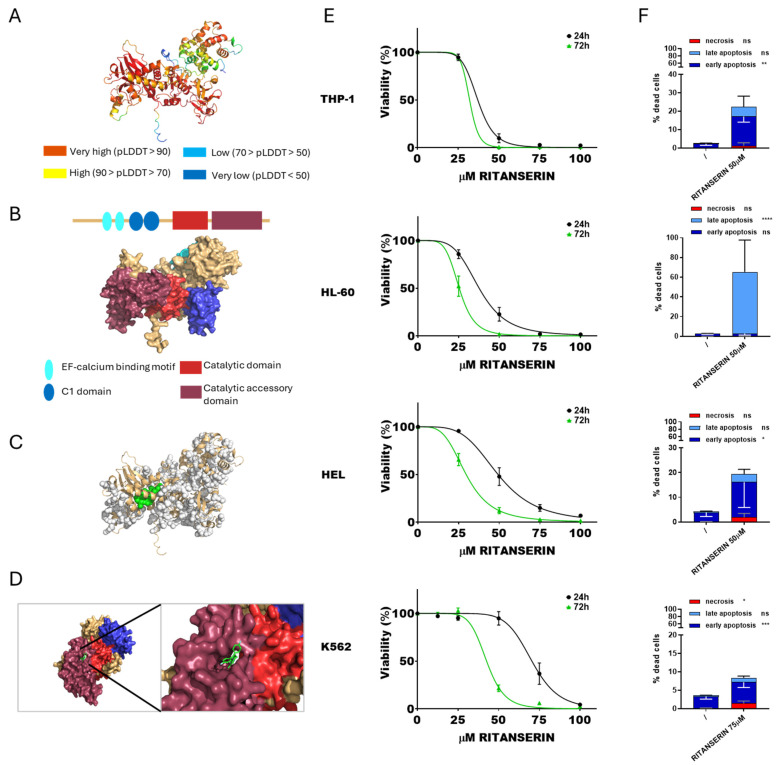
DGKα and ritanserin: from simulation to effects on AML cell viability. (**A**) Predicted DGKα structure generated by AlphaFold. (**B**) Schematic reconstruction of DGKα principal functional domains. (**C**) Predicted DGKα binding pockets. Pocket 56 is highlighted in green. (**D**) Visualization of the interaction between DGKα pocket 56 and ritanserin. (**E**) AML cells were treated with increasing concentration of ritanserin for 24 or 72 h followed by alamarBlue viability assay for an additional 24 h. Experiments were performed in quadruplicates. Data represent mean ± SEM of six or more experiments interpolated as [inhibitor] vs. % viability. (**F**) AML cells were treated with concentrations of ritanserin near the previously determined IC_50_ values and analyzed by flow cytometry to detect necrotic (Annexin− and 7AAD+), early apoptotic (Annexin V+ and 7AAD−), and late apoptotic (Annexin+ and 7AAD+) cell populations. Data represent mean ± SD of three or more independent experiments. Statistical analysis was performed using two-way ANOVA: *p* < 0.05 *, *p* < 0.01 **, *p* < 0.001 ***, *p* < 0.001 **** and ns *p* not significant.

**Figure 2 cells-14-01721-f002:**
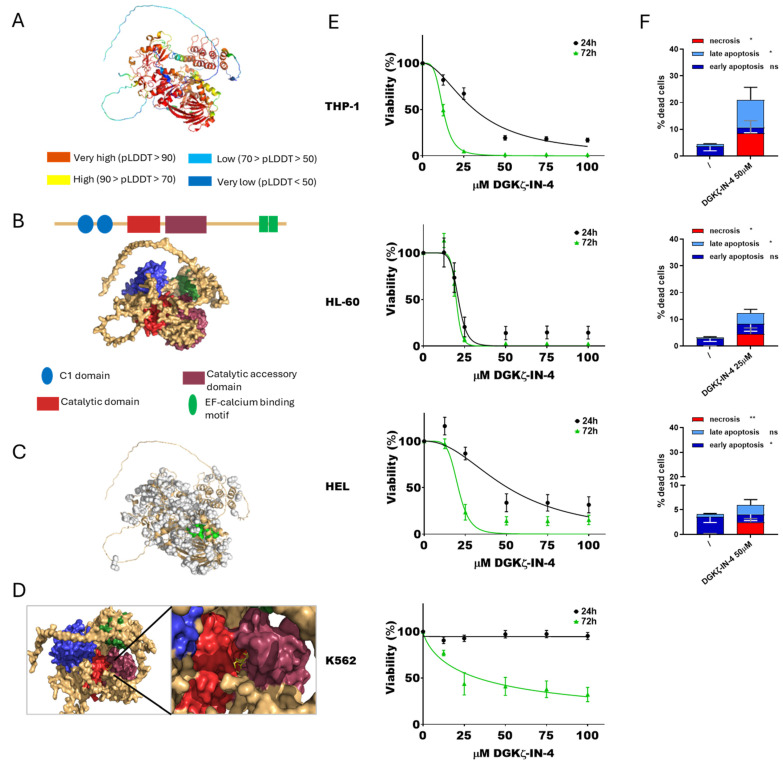
DGKζ and DGKζ-IN-4: from simulation to effects on AML cell viability. (**A**) Predicted DGKζ structure generated by AlphaFold. (**B**) Schematic reconstruction of DGKζ principal functional domains. (**C**) Predicted DGKζ binding pockets. Pocket 65 in shown green. (**D**) Visualization of the interaction between DGKζ pocket 65 and DGKζ-IN-4. (**E**) AML cells were treated with increasing concentration of DGKζ-IN-4 for 24 or 72 h followed by alamarBlue viability assay for an additional 24 h. Experiments were conducted in quadruplicates. Data are shown as the mean ± SEM of five or more experiments interpolated as [inhibitor] vs. % viability. (**F**) AML cells were treated with concentrations of DGKζ-IN-4 near the previously determined IC_50_ values and analyzed by flow cytometry to detect necrotic (Annexin− and 7AAD+), early apoptotic (Annexin V+ and 7AAD−), and late apoptotic (Annexin+ and 7AAD+) cell populations. Data represent mean ± SD of three or more independent experiments presenting the percentage of dead cells. Statistical analysis was performed using two-way ANOVA test, *p* < 0.05 *, *p* < 0.01 ** and ns *p* not significant.

**Figure 3 cells-14-01721-f003:**
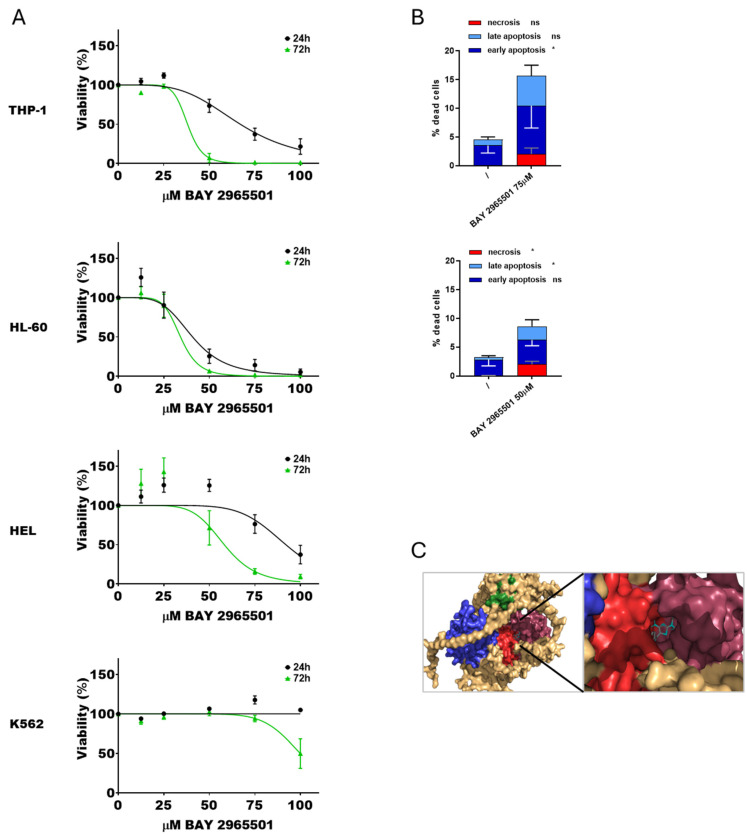
Effects of BAY 2965501 on AML cell viability. (**A**) AML cells were treated with increasing concentration of BAY 2965501 for 24 or 72 h, followed by alamarBlue viability assay for an additional 24 h. Experiments were performed in quadruplicates. Data are shown as mean ± SEM of five or more experiments interpolated as [inhibitor] vs. % viability. (**B**) AML cells were treated with concentrations of BAY 2965501 near the previously obtained IC_50_ values and analyzed by flow cytometry to detect necrotic (Annexin− and 7AAD+), early apoptotic (Annexin V+ and 7AAD−), and late apoptotic (Annexin+ and 7AAD+) cell populations. Data represent mean ± SD of three or more independent experiments showing the percentage of dead cells. Statistical analysis was performed using two-way ANOVA test, *p* < 0.05 * and ns *p* not significant. (**C**) Visualization of the interaction between DGKζ pocket 65 and BAY 2965501.

**Figure 4 cells-14-01721-f004:**
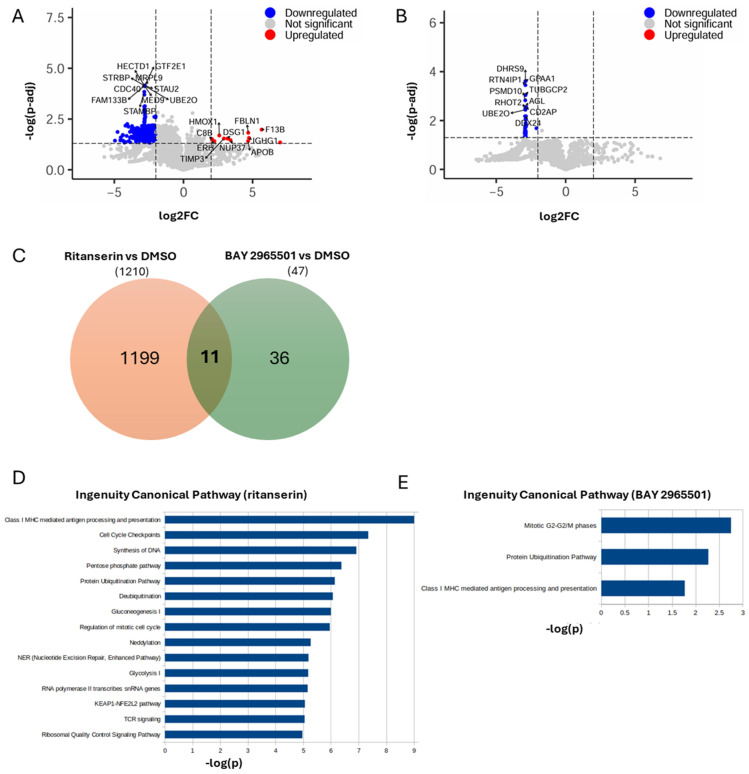
Proteomic changes induced by ritanserin and BAY 2965501. THP-1 cells were treated with ritanserin (50 µM) and BAY 2965501 (75 µM). After 24 h, proteins were extracted and quantified by mass spectrometry. (**A**) Volcano plot showing differentially abundant proteins between ritanserin and DMSO. Upregulated proteins are highlighted in red, downregulated proteins in blue. For both categories, the names of the top 10 most significantly up- or downregulated proteins are shown. (**B**) Volcano plot showing differentially abundant proteins between BAY 2965501 and DMSO. Only down-regulated proteins are present (blue), with the names of the top 10 most significant indicated. (**C**) Venn diagram representing the overlap between differentially abundant proteins in ritanserin vs. DMSO (orange) and BAY vs. DMSO (green). (**D**) Bar plot of significant pathways obtained with IPA using as input differentially abundant proteins between ritanserin and DMSO. Blue bars indicate that all pathways are downregulated. (**E**) Bar plot of significant pathways obtained with IPA using as input differentially abundant proteins between BAY 2965501 and DMSO. Blue bars indicate downregulated pathways.

**Figure 5 cells-14-01721-f005:**
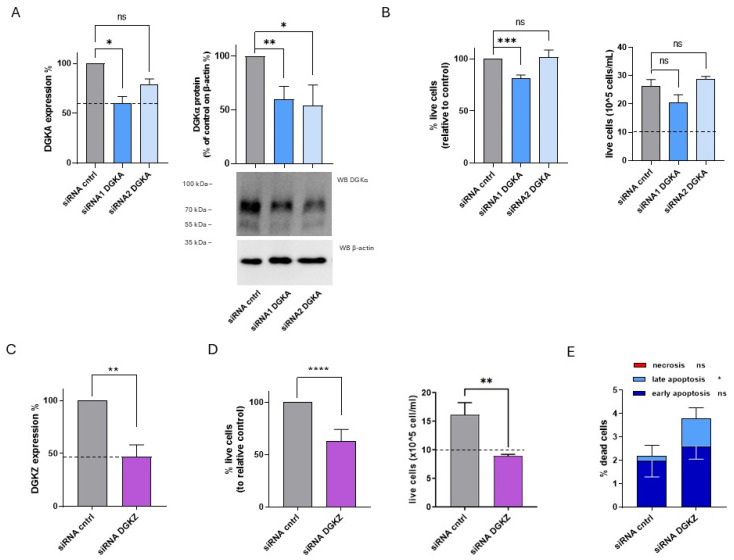
Role of DGKA and DGKZ in HL-60 viability. HL-60 cells were transfected with the indicated siRNAs and assessed after 72 h. (**A**) Mean ± SEM of three independent experiments showing the silencing of DGKA in HL-60 cells. Data are presented as percentage of DGKA mRNA expression (**left**) or DGKA protein (**right**) in comparison with siRNA control (cntrl)-transfected cells. Bottom right: representative Western Blot for DGKA and β-actin. (**B**) Trypan blue assay. Mean ± SEM of four independent experiments showing the percentage of live cells relative to siRNA cntrl (**left**). Right: representative experiment showing absolute numbers of live cells starting from the 10 × 10^5^ cells/mL seeded. (**C**) Mean ± SEM of three independent experiments showing the silencing of DGKZ in HL-60 cells, expressed as percentage of DGKZ mRNA expression relative to siRNA cntrl-transfected cells. (**D**) Trypan blue assay. Mean ± SEM of seven independent experiments showing the percentage of live cells relative to siRNA cntrl (**left**). Right: representative experiment showing absolute numbers of live cells starting from the 10 × 10^5^ cells/mL seeded. (**E**) Annexin V-7AAD assay. Data represent the mean ± SD of five independent experiments showing the percentage of dead cells and the type of cell death. Statistical significance was determined using two-way ANOVA or unpaired *t* test vs. siRNA cntrl: *p* < 0.05 *, *p* < 0.01 **, *p* < 0.001 ***, *p* < 0.0001 **** and ns *p* not significant.

**Figure 6 cells-14-01721-f006:**
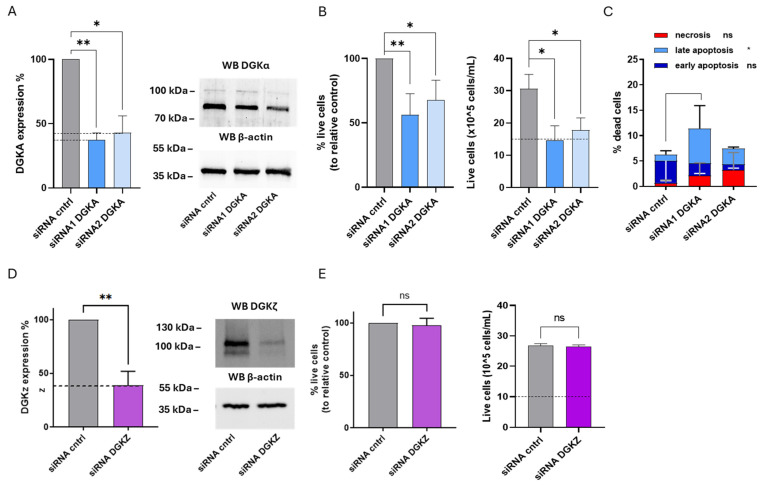
Role of DGKα and DGKζ in HEL viability. HEL cells were transfected with the indicated siRNA and assessed after 72 h. (**A**) Mean ± SEM of three independent experiments showing the silencing of DGKA in HEL cells. Data are presented as percentage of DGKA mRNA expression (**left**) in comparison with siRNA cntrl-transfected cells. Right: representative Western Blot for DGKA and β-actin. (**B**) Trypan blue assay. Mean ± SEM of four independent experiments showing the percentage of live cells relative to siRNA cntrl (**left**). Right: representative experiment showing absolute numbers of live cells starting from 10 × 10^5^ cells/mL seeded. (**C**) Annexin V-7AAD assay. Data represent mean ± SD of three independent experiments showing the percentage of dead cells and the type of cell death. (**D**) Mean ± SEM of three independent experiments showing DGKZ silencing in HEL cells. Data are presented as percentage of DGKZ mRNA expression in comparison with siRNA cntrl-transfected cells (**left**). Right, a representative Western Blot for DGKZ and β-actin protein. (**E**) Trypan blue assay. Mean ± SEM of four independent experiments showing the percentage of live cells relative to siRNA cntrl (**left**). Right: representative experiment showing absolute numbers of live cells starting from 10 × 10^5^ cells/mL seeded. Statistical significance was determined through two-way ANOVA or unpaired *t* test vs. siRNA cntrl: *p* < 0.05 *, *p* < 0.01 ** and ns *p* not significant.

**Figure 7 cells-14-01721-f007:**
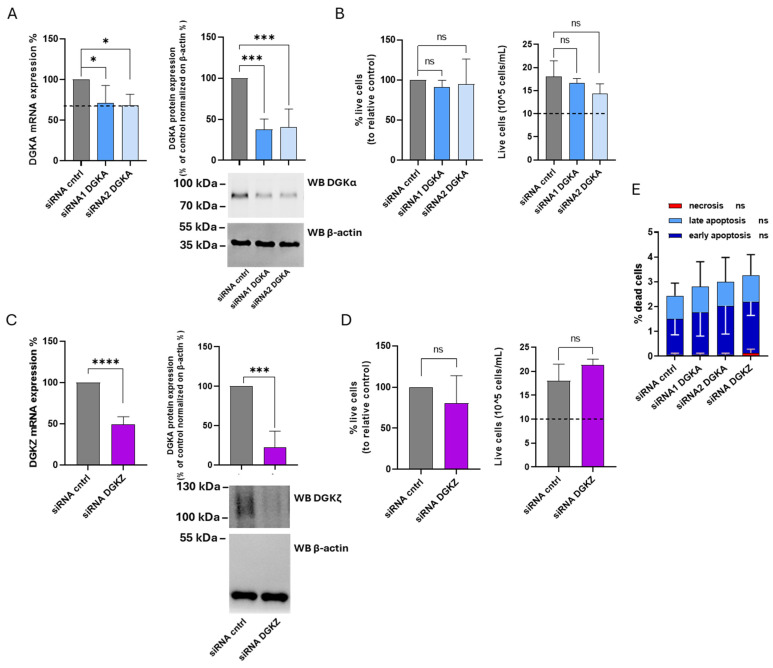
Role of DGKα and DGKζ in THP-1 viability. THP-1 cells were transfected with the indicated siRNAs and assessed after 72 h. (**A**) Mean ± SEM of three independent experiments showing the silencing of DGKA in THP-1 cells. Data are shown as percentage of DGKA mRNA expression (**left**) or DGKA protein (**right**) ± SEM in comparison with siRNA cntrl-transfected cells. Bottom right: representative Western Blot for DGKα and β-actin protein. (**B**) Trypan blue assay. Mean ± SEM of four independent experiments showing the percentage of live cells relative to siRNA cntrl (**left**). Right: representative experiment showing the number of live cells starting from 10 × 10^5^ cells/mL seeded. (**C**) Mean ± SEM of three independent experiments showing the silencing of DGKZ in THP-1 cells. Data are presented as percentage of DGKZ mRNA expression (**left**) or DGKZ protein (**right**) in comparison with siRNA cntrl-transfected cells. Lower right, a representative Western Blot for DGKζ and β-actin protein. (**D**) Trypan blue assay. Mean ± SEM of four independent experiments showing the percentage of live cells relative to siRNA cntrl (**left**). Right: representative experiment showing the number of live cells starting from 10 × 10^5^ cells/mL seeded. (**E**) Annexin V-7AAD assay. Data are expressed as mean ± SD of three independent experiments showing the percentage of dead cells and the type of cell death. The significance was determined through two-way ANOVA or unpaired *t*-test vs. siRNA cntrl: *p* < 0.05 *, *p* < 0.001 ***, *p* < 0.0001 **** and ns *p* not significant.

**Figure 8 cells-14-01721-f008:**
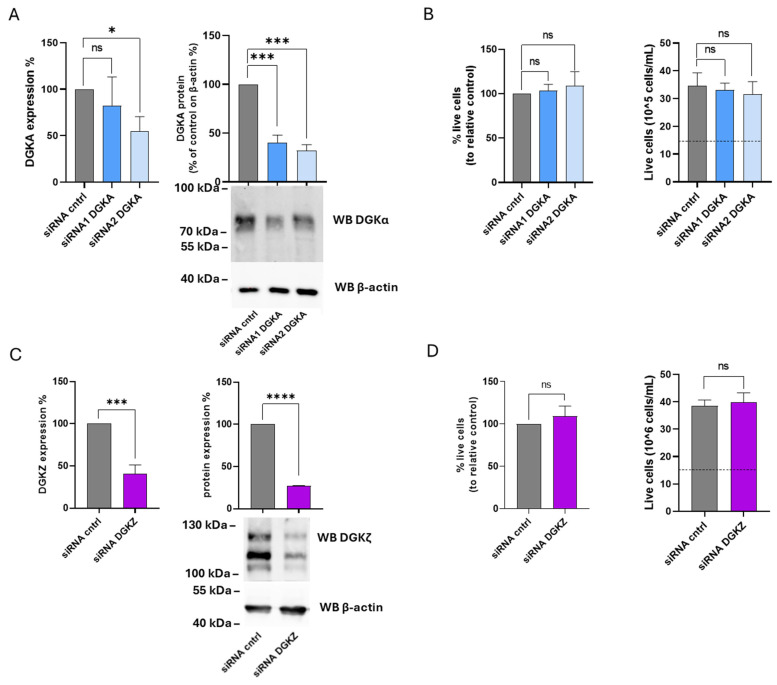
Role of DGKA and DGKZ in K562 viability. K562 cells were transfected with the indicated siRNAs and assessed after 72 h. (**A**) Mean ± SEM of three independent experiments showing the silencing of DGKA in K562 cells. Data are expressed as percentage of DGKA mRNA expression (**left**) or DGKA protein (right) in comparison with siRNA cntrl-transfected cells. Lower right: representative Western Blot for DGKA and β-actin. (**B**) Trypan blue assay. Mean ± SEM of three independent experiments showing the percentage of live cells relative to siRNA cntrl (**left**). Right: representative experiment showing absolute numbers of live cells starting from 15 × 10^5^ cells/mL seeded (dashed line). (**C**) Mean ± SEM of three independent experiments showing the silencing of DGKZ in K562 cells. Data are presented as a percentage of DGKZ mRNA expression (**left**) or DGKZ protein (**right**) in comparison with siRNA cntrl-transfected cells. Lower right, a representative Western Blot for DGKZ and β-actin protein. (**D**) Trypan blue assay. Mean ± SEM of three independent experiments showing the percentage of live cells relative to siRNA cntrl (**left**). Right: representative experiment showing the number of live cells starting from 15 × 10^5^ cells/mL seeded. The significance was assessed through two-way ANOVA or unpaired *t*-test vs. siRNA cntrl: *p* < 0.05 *, *p* < 0.001 ***, *p* < 0.0001 **** and ns *p* not significant.

**Table 1 cells-14-01721-t001:** Electroporation conditions.

	Condition	N° of Cells	Vitality (%)	Efficiency (%)
HL-60	1300 V, 35 ms, 1 P	500,000		
HEL	1350 V, 20 ms, 2 P	760,000	38	100
THP-1	1550 V, 15 ms, 2 P	750,000	89	81
K562	1350 V, 10 ms, 4 P	1,500,000	87	71

## Data Availability

Proteomic data are available via ProteomeXchange with identifier PXD069537. Further original contributions presented in this study are included in the article/Appendix A. Inquiries can be directed to the corresponding author.

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
