# Peer review of "Acute Myeloid Leukemia: A Key Role of DGKα and DGKζ in Cell Viability"

_cells, 2025, doi:10.3390/cells14211721_

Round 1

Reviewer 1 Report

Comments and Suggestions for Authors

1.This research focused on Acute Myeloid Leukemia: a key role of DGKα and DGKζ in cell viability, after check in pubmed, to be fact no many references about this topic, so this manuscript with some importance and novelty.

2.Whole manuscript contained so much data and contents, but I think some places can be more perfect.

  1. Graphical Abstract can add if the journal needed.

4.Only cell research,no animal or clinical research restricted the depth and height of this article.

  1. In Materials and Methods section,there is no need to list all the reagents and materials.
  2. Why try electroporated not Chemical transfection such as Lipo.
  3. Figures need revised completely such as Figure 1 F necrosis red cube not same, AND Figures was ABCDEF but figure legends was abcdef.
  4. So many WB results, please support original film image.

Author Response

  1. This research focused on Acute Myeloid Leukemia: a key role of DGKα and DGKζ in cell viability, after check in pubmed, to be fact no many references about this topic, so this manuscript with some importance and novelty.

Thank you for your appreciation of our work and its novelty.

2.Whole manuscript contained so much data and contents, but I think some places can be more perfect.

We revised the manuscript to improve its quality according to your and other revisors’ comments. We also had expert English editing.

  1. Graphical Abstract can add if the journal needed.

We added a graphical abstract.

4.Only cell research,no animal or clinical research restricted the depth and height of this article.

Indeed, the present study is limited to commonly used human cell lines and is preliminary to further research in animal models/patient derived cells. This is now better explained in conclusions.

  1. In Materials and Methods section,there is no need to list all the reagents and materials.

We moved the list to supplementary materials.

  1. Why try electroporated not Chemical transfection such as Lipo.

In our experience chemical transfection has very limited efficiency on those cells. In pilot experiments only a minority of cell were transfected by lipofectamine. Therefore, we decided to optimize electroporation.

  1. Figures need revised completely such as Figure 1 F necrosis red cube not same, AND Figures was ABCDEF but figure legends was abcdef.

We revised all the cubes to have similar dimensions and corrected capital letters in the legend.

  1. So many WB results, please support original film image.

Original images were provided to the editorial staff and are now also in supplementary materials (Supplementary Data 3: uncropped western blots).

Reviewer 2 Report

Comments and Suggestions for Authors

This study is interesting and the manuscript is well organized.

Comments:

  1. Some keywords have too broad meanings, replace specific keywords related to this research.
  2. Table 1-3, Please use the standardized three line table and refer to the journal submission requirements.
  3. The description of the results does not require citing references, only describing the findings of this study, and discussing in detail the comparison of the results with other studies.
  4. The caption of Figure 3: Lack of explanation for C diagram.
  5. Add content to the conclusion section, summarize existing results, and clarify shortcomings and future research directions.
  6. Abbreviations do not need to be presented in a table format, they can be listed directly. Please refer to the submission guidelines and templates.
  7. The raw data of proteomics needs to be uploaded to public databases such as NCBI, and the link and login number after successful data upload should be provided in the Data Availability Statement section.
  8. The original gel images of WB cannot show the results of other duplicate samples.

Author Response

This study is interesting and the manuscript is well organized.

Thank you for your appreciation of our work.

Comments:

  1. Some keywords have too broad meanings, replace specific keywords related to this research.

We revised keyword narrowing subject as suggested.

Keywords: Acute myeloid leukemia; diacylglycerol kinases inhibition; diacylglycerol kinase isoform silencing; drug sensitivity; proteome remodeling; lipid signaling.

  1. Table 1-3, Please use the standardized three line table and refer to the journal submission requirements.

All the tables were reformatted according to journal guidelines. Please note, we moved table 1 and table 2 to supplementary materials as suggested by reviewer 1.

  1. The description of the results does not require citing references, only describing the findings of this study, and discussing in detail the comparison of the results with other studies.

As suggested, we moved introductory text and reference to the introduction, maintaining only the strictly necessary citations in results section.

  1. The caption of Figure 3: Lack of explanation for C diagram.

We revised the legend accordingly.

Figure 3. Effects of BAY 2965501 on AML cell viability. (A) AML cells were treated with increasing concentration of BAY 2965501 for 24 or 72 h, followed by alamarBlue viability assay for an additional 24 h. Experiments were performed in quadruplicates. Data are shown as mean ± SEM of five or more experiments interpolated as [inhibitor] vs. % viability. (B) AML cells were treated with concentrations of BAY 2965501 near the previously obtained IC50 values and analyzed by flow cytometry to detect necrotic (Annexin- and 7AAD+), early apoptotic (Annexin V+ and 7AAD-), and late apoptotic (Annexin+ and 7AAD+) cell populations. Data represent mean ± SD of three or more independent experiments showing the percentage of dead cells. Statistical analysis was performed using two-way ANOVA test, p < 0.05*. (C) Visualization of the interaction between DGKζ pocket 65 and BAY 2965501.

  1. Add content to the conclusion section, summarize existing results, and clarify shortcomings and future research directions.

We added a conclusions section with suggested content.

  1. Conclusion

Targeting DGKα and DGKζ effectively reduces AML cell viability, with both isoforms exhibiting distinct and cell type-specific functional roles. However, the observed drug sensitivity does not correlate with DGKα or DGKζ expression levels, indicating that additional cellular or molecular factors may modulate responses to these inhibitors. The substantial heterogeneity of AML, together with the limited selectivity of current DGK inhibitors, underscores the importance of identifying predictive biomarkers of sensitivity to DGKα or DGKζ blockade, which will be essential for defining responsive patient subgroups and guiding future clinical applications. Further studies employing patient-derived AML samples and in vivo models will be crucial to translate DGK inhibition into an effective therapeutic strategy.

  1. Abbreviations do not need to be presented in a table format, they can be listed directly. Please refer to the submission guidelines and templates.

We revised the abbreviations list accordingly.

  1. The raw data of proteomics needs to be uploaded to public databases such as NCBI, and the link and login number after successful data upload should be provided in the Data Availability Statement section.

Proteomic data are now available via ProteomeXchange with identifier PXD069537.

Submission details:

 Project Name: Acute Myeloid Leukemia: a key role of DGKα and DGKζ in cell viability.

 Project accession: PXD069537

Reviewer access details

Username: reviewer_pxd069537@ebi.ac.uk

 Password: ePy0ikx4lpZf

  1. The original gel images of WB cannot show the results of other duplicate samples.

All WB was carefully revised and uncropped figures are now in supplementary materials (Supplementary Data 3: uncropped western blots)..

Reviewer 3 Report

Comments and Suggestions for Authors

In this study, the effect of ritanserin, a DGKα specific inhibitor and DGKζ-IN4 / BAY 2965501, DGKζ specific inhibitor, on a panel of acute myeloid leukemia (AML) cell models was investigated. The effects observed with isoform specific silencing and finally explored the effect of those drugs on THP-1 cell proteome were compared. Although it is a relatively comprehensive study and some positive results were obtained, there are only cell models tested. Followings are some suggestions for revisions.

  1. Please explain in the text why they chose those cell models.
  2. It is suggested to further study on the animal models for AML.
  3. More key results can be provided in the abstract.
  4. The significant and novelty of present work can be emphasized in the last paragraph of introduction section. A schematic diagram of present work may be provided here.
  5. It is better not to use the first-person “we” narrative for a scientific paper.
  6. Please use three-line format tables.
  7. There should be a space between the numeral and unit, including in the text and figures; the “hours” can be changed to “h”; the “m/z” should be in Italics font.
  8. How to determine the dosage, such as “12.5 to 100 μM” (line 339).
  9. A separated conclusion section can be added.
  10. As it is indicated that “the aim of our study is to extend the fragmented data present in the literature……”, a schematic diagram of the action mechanism can be provided in the discussion section.

Author Response

In this study, the effect of ritanserin, a DGKα specific inhibitor and DGKζ-IN4 / BAY 2965501, DGKζ specific inhibitor, on a panel of acute myeloid leukemia (AML) cell models was investigated. The effects observed with isoform specific silencing and finally explored the effect of those drugs on THP-1 cell proteome were compared. Although it is a relatively comprehensive study and some positive results were obtained, there are only cell models tested.

Thank you for your appreciation of our work.

Followings are some suggestions for revisions.

  1. Please explain in the text why they chose those cell models.

We describe in the introduction the reasons for our choice of cell lines.

To reflect AML heterogeneity, four distinct models were selected. HL-60 cells, derived from acute promyelocytic leukaemia [16], were chosen due to their strong differentiation potential along the myeloid lineage [17-21]. HEL cells, established from a case of erythroleukemia, were included for their erythroid characteristics and presence of relevant driver mutations, making them suitable for modeling pure erythroid leukaemia [22]. THP-1 cells, obtained from acute monocytic leukaemia, harbor the typical t(9;11) translocation and reproduce chemoresistant and invasive phenotypes [23]. Upon stimulation they can differentiate into macrophage-like cells [24], providing a model to investigate DGK functions in monocyte-to-macrophage transition. Finally, K562 cells, although originally derived from chronic myeloid leukemia (CML), were included because of their frequent use in AML research owing to their conserved myeloid properties and ability to differentiate towards erythroid or megakaryocytic lineages under specific treatments [25-27].

  1. It is suggested to further study on the animal models for AML.

Indeed, the present study is limited to commonly used human cell lines and is preliminary to further research in animal models/patient derived cells. This is now better explained in the conclusions section.

  1. More key results can be provided in the abstract.

Abstract was revised to include more data and improve clarity.

Abstract

Acute myeloid leukemia (AML) is a heterogeneous disease with an unmet need for novel therapeutic drugs. Previous studies have reported the upregulation of diacylglycerol kinases (DGKs) in AML. This study investigated the effects of ritanserin, a DGKα-specific inhibitor, and DGKζ-IN4 or BAY 2965501, DGKζ-selective inhibitors, on a panel of AML cell lines. Ritanserin induced apoptotic cell death across all tested models, whereas DGKζ inhibitors triggered both apoptosis and necrosis to variable extents, with HL-60 cells being the most responsive to both compounds. Drug sensitivity did not correlate with DGKα or DGKζ expression levels, indicating that additional factors may influence cellular susceptibility. THP-1 proteomic profiling revealed that ritanserin broadly downregulated proteins involved in antigen presentation, cell cycle and metabolism, while BAY 2965501 affected a smaller and distinct but functionally similar protein subset, implying different mechanisms of action. Gene silencing confirmed AML cell line-specific dependence on DGK isoforms: HEL cells were sensitive to DGKα knockdown, HL-60 to DGKζ silencing, whereas K562 and THP-1 were resistant to both. These findings indicate that DGKs targeting can effectively reduce AML cell viability. However, AML heterogeneity and the limited selectivity of current inhibitors underscore the need for predictive biomarkers and combinatorial strategies to translate DGK inhibition into effective therapy.

  1. The significant and novelty of present work can be emphasized in the last paragraph of introduction section. A schematic diagram of present work may be provided here.

We extensively revised the last paragraph of the introduction and added a graphical abstract to the paper.

  1. It is better not to use the first-person “we” narrative for a scientific paper.

Professional English revision was done on the full manuscript taking this indication in consideration.

  1. Please use three-line format tables.

All the tables were reformatted according to journal guidelines. Please note, we moved table 1 and table 2 to supplementary materials as suggested by reviewer 1.

  1. There should be a space between the numeral and unit, including in the text and figures; the “hours” can be changed to “h”; the “m/z” should be in Italics font.

Done accordingly.

  1. How to determine the dosage, such as “12.5 to 100 μM” (line 339).

Dosage and the number of experimental points were chosen based on preliminary experiments in order to cover from no effect to full cell death.

  1. A separated conclusion section can be added.

We added a conclusion section.

  1. Conclusion

Targeting DGKα and DGKζ effectively reduces AML cell viability, with both isoforms exhibiting distinct and cell type-specific functional roles. However, the observed drug sensitivity does not correlate with DGKα or DGKζ expression levels, indicating that additional cellular or molecular factors may modulate responses to these inhibitors. The substantial heterogeneity of AML, together with the limited selectivity of current DGK inhibitors, underscores the importance of identifying predictive biomarkers of sensitivity to DGKα or DGKζ blockade, which will be essential for defining responsive patient subgroups and guiding future clinical applications. Further studies employing patient-derived AML samples and in vivo models will be crucial to translate DGK inhibition into an effective therapeutic strategy.

  1. As it is indicated that “the aim of our study is to extend the fragmented data present in the literature……”, a schematic diagram of the action mechanism can be provided in the discussion section.

We preferred to add a graphical abstract. As this paper has already many figures (each with multiple panels) further graphs can make it hard to follow.

Reviewer 4 Report

Comments and Suggestions for Authors

The article is an interesting analysis of diacylglycerol kinases in myeloid leukemia. The rationale, the methodology and the presentation of the results is adequate. The authors have provided a detailed analysis of the different drugs involved and the mechanism. The figures and table are informative and the discussion is centered in the results and provides an important background for future research. I have only two minor suggestions. One refers to the use of the cell line K562 and its partial resistance to the compounds used. It would be interesting to develop cell lines resistant to the compounds in order to define the possible mutation sites involved in resistance. The other issue is the analysis of primery culture. These enzymes are also crucial for several physiological responses. Probably a good delivery system to enhance the uptake in malignant cells to increase drug efficiency should be envisioned in future studies. 

Author Response

The article is an interesting analysis of diacylglycerol kinases in myeloid leukemia. The rationale, the methodology and the presentation of the results is adequate. The authors have provided a detailed analysis of the different drugs involved and the mechanism. The figures and table are informative and the discussion is centered in the results and provides an important background for future research.

Thank you for your appreciation of our work.

I have only two minor suggestions. One refers to the use of the cell line K562 and its partial resistance to the compounds used. It would be interesting to develop cell lines resistant to the compounds in order to define the possible mutation sites involved in resistance.

The other issue is the analysis of primery culture. These enzymes are also crucial for several physiological responses. Probably a good delivery system to enhance the uptake in malignant cells to increase drug efficiency should be envisioned in future studies. 

All of those are interesting suggestions now included in the discussion for future developments.

Round 2

Reviewer 1 Report

Comments and Suggestions for Authors

The authors have revised completely all my concerns,can be accepted.

Reviewer 2 Report

Comments and Suggestions for Authors

The authors have addressed all my comments, it can be accepted in the current form.

Reviewer 3 Report

Comments and Suggestions for Authors

All the comments have been well-addressed, and the manuscript has been carefully revised accordingly. This manuscript may be accepted for publication in current form. During proof reading, pleas pay attention to some minor issues. 1, The Graphical Abstract (a visual representation that combines images and brief text to summarize the essential content and innovations of a research paper) has not been found in the manuscript. 2, The resolution of figures can be further improved. 3, There should be a space between the numeral and unit, such as in the Figure 1-3.